# A Health Equity Implementation Approach to Child Health Literacy Interventions

**DOI:** 10.3390/children9091284

**Published:** 2022-08-25

**Authors:** Lucio Naccarella, Shuaijun Guo

**Affiliations:** 1Melbourne School of Population and Global Health, University of Melbourne, Melbourne, VIC 3053, Australia; 2Centre for Community Child Health, Murdoch Children’s Research Institute, Royal Children’s Hospital, Melbourne, VIC 3052, Australia; 3Department of Paediatrics, University of Melbourne, Melbourne, VIC 3052, Australia

**Keywords:** health literacy, health equity, children, implementation, interventions

## Abstract

Health and behavioural inequalities exist in all populations, including children. As a social determinant of health, health literacy is a crucial driver of equitable health outcomes in children. With the increasing calls for more actions on addressing low health literacy and inequalities, health literacy interventions to improve children’s healthy behaviours have emerged as a key strategy to reduce health inequities. However, health literacy interventions face implementation challenges impacting upon potential outcomes, and disparities in the implementation of health literacy interventions also occur. Variation exists in child health literacy intervention target groups, timing, content and formats, and there is a lack of implementation specificity, resulting in a lack of clarity about which intervention strategies are the most effective in improving health literacy, related health behaviours, and associated health outcomes. While actions to facilitate child health intervention implementation exist, to minimise further perpetuation of child health inequities, this perspective calls for a health equity implementation approach to child health literacy interventions.

## 1. Introduction

Health and behavioural inequalities exist in all populations, including children. Disparities in child health literacy have been documented between high-income and low- and middle-income countries, as evidenced by child health literacy being distributed unequally across sociodemographic groups, confirming a social gradient in low health literacy [1,2,3,4]. While the field of child health literacy is in its early stages, improving child health literacy is an important means of decreasing health inequities, as it can be changed by education and learning to reduce avoidable health disparities [5,6,7].

As a social determinant of health, health literacy is a crucial driver of equitable health outcomes in children [8,9]. For example, knowledge is a key component of health literacy in children. Through education and learning, children gain knowledge enabling them to approach health information competently and effectively and to make health-promoting decisions and actions [10].

With the increasing calls for more actions on addressing health literacy and inequalities [10], child health literacy interventions have emerged to bring about improvements in healthy behaviours [11]. School-based health literacy interventions are ideally positioned to address the low health literacy of children, with potential to mitigate children’s health disparities, as they allow for early and ongoing interventions [3,12,13,14]. The education system, and specifically schools, are widely recognised as a key setting for child health literacy interventions [6]. Child health literacy interventions can include health literacy education in schools and classrooms for students, as well as teacher professional education and training [15]. Key mechanisms through which interventions lead to improved healthy behaviours can include: the school curriculum, healthy school environment and engagement with families and communities [7]. Key strengths and challenges of schools as settings for child health literacy interventions are also known (Table 1). While there is a recognised need for interventions to include implementation fidelity evaluation, there is a lack of attention to assessing the equity implementation context of school-based health literacy interventions.

Child health literacy interventions have been found to have variability in target groups, timing, content and formats, and a lack of implementation specificity, resulting in a lack of clarity about which interventions are the most effective in improving health literacy, related health behaviours, and associated health outcomes [18]. Given the complexity of health literacy interventions and the settings within which they are implemented, it is not surprising that implementation challenges also exist. For example, in relation to Building Wellness (as a Youth Health Promotion Program), Diamond reported [19] that the “the variability in school pilot site practices and policies regarding health education and the variety in program facilitators impacted upon implementation and program outcomes”. In making the case for child health literacy interventions, the World Health Organisation (Geneva, Switzerland) [3] also pointed to the challenges to implementation and advocated key implementation specific actions, including:Ensuring health literacy is an integral element of teacher training and school curriculums;Developing supportive strategies for implementation at the individual school levelUsing regulatory mechanisms to allow pooling of funding and responsibility for policy and programmes between health and education sectors;Developing financial and reputational incentive schemes to promote fidelity in the implementation of education-system based health literacy programmes;Tailoring health literacy programmes to the local education sector context;Strengthening the evidence base on effectiveness and cost-effectiveness from an education, as well as health system, perspective.

While actions to facilitate child health interventions implementation exist [20], to minimise further perpetuation of child health inequities, with the emergence of implementation science, calls have emerged for a health equity implementation approach [21].

## 2. Health Equity and Implementation

Health equity means that no one is denied the possibility to have optimal health and wellbeing, even if belonging to a group that has been or is economically or socially disadvantaged [22]. Interventions can perpetuate disparities in health for particular population groups, if their implementation results in worse access, receipt, use, quality or outcomes. Evidence exists that race, ethnicity, sexual orientation, gender identity, socioeconomic status, functional limitations, and other characteristics can contribute to disparities in the implementation of health interventions. [22]. With the increasing focus on bridging the gap between evidence and practice, there is now recognition that researchers and practitioners should use implementation determinant frameworks to understand why disparities in health implementation occur [23].

## 3. Implementation Science, Intervention Implementation and Health Literacy

The role and importance of implementation science as the study of strategies to promote the uptake of research into real-world settings is established [24]. The role and relationship between health literacy and intervention implementation is multi-faceted and under-researched. A recent review [25] revealed three key ways in which health literacy features in intervention implementation: (1) in designing and developing interventions, (2) as a contextual factor influencing implementation success, and (3) as an outcome the healthcare intervention [25]. Overall, however, there is a lack of recognition of the importance of health equity informed implementation of health literacy interventions, and the role that implementation science in facilitating health literacy is under-researched.

Within implementation science, three core types of implementation science frameworks exist [26]: (1)Determinant frameworks—to identify and establish what factors determine or predict implementation success;(2)Process frameworks—to clarify how to address determinants to achieve implementation success;(3)Evaluation frameworks—to determine the metrics and assessment to identify implementation success.

A Health Equity Implementation Framework [21] has been developed and recommends the integration of three health equity domains into implementation science frameworks, namely: (1)Healthcare intervention recipient cultural factors—for example, socioeconomic status, race and/or ethnicity, language;(2)Interaction between Child and Educator—for example, interactions can predict satisfaction, perceived trust, and health outcomes;(3)Societal context—that may relate to economic, demographic, or geographical factors. 

While several implementation frameworks incorporate an equity lens (e.g., RE-AIM [27]), no specific child health literacy equity implementation frameworks exist. 

## 4. Child Health Literacy Interventions with an Equity Focus 

While the need to adapt child health literacy interventions to local contexts is recognised in practice (e.g., HealthLit4Kids, HeLit-Schools) [13,28], a gap remains in using principles of health equity for the implementation of child health literacy interventions. Implementation determinant frameworks need to be adapted to identify and understand barriers to equitable implementation of child health literacy interventions to inform interventions designs and optimal implementation strategies [29].

In this perspective, we use the Health Equity Implementation Framework [21] to illustrate a health equity approach to the implementation of child health literacy interventions. The three health equity domains have been integrated into the Consolidated Framework for Implementation Research (CFIR) [20]. The CFIR is a widely used implementation determinant framework to facilitate design, implementation and evaluation of evidence-based interventions in child health in multiple healthcare [30,31] and non-healthcare (e.g., schools [32,33]) settings. CFIR has five domains—Intervention characteristics, Inner Context, Outer Context, Recipients, and Process—that allow the comprehensive identification and understanding of factors that many determine or predict implementation success. Table 2 provides illustrative questions for researchers and policy makers to guide the design and evaluation of equitable implementation of child health literacy interventions.

## 5. Conclusions

While actions to facilitate child health literacy intervention implementation exist, to minimise the further perpetuation of child health literacy inequities, researchers and practitioners are encouraged to adopt a health equity lens as a foundation to better inform the design, adoption, implementation and evaluation of equitable child health literacy interventions.

## Figures and Tables

**Table 1 children-09-01284-t001:** Strengths and Challenges of School-based Child Health Literacy interventions.

Strengths	Challenges
Schools are accessible to all; access to age appropriate health education exists; opportunities exist for all children to develop higher levels of health literacy; opportunities exist to develop lifelong learning skills; access exists to free health information [6] Health literacy education embedded in school curriculum; building healthy school environments using the Health Promoting Schools (HPS) Framework, promote critical health literacy [15,16,17]/> Comprehensive integrated approaches can be used that target individuals attitudes and behaviours, as well as the school environment [17]	Health literacy is not well known in the education sector; health education and promotion is not part of the school curriculum; health is not part of core goals of education [15] Predominantly targeted at 13–18 year old adolescents; hence, too late to influence health behaviours; limited program planning or measurement tools for implementation and evaluation; limited use of whole of school approaches; lack of teacher preparedness, confidence and institutional time; inadequate and limited in-service training opportunities for teachers [13] Variable implementation fidelity of health literacy programs and the original HPS framework; limited detailing of health literacy outcomes [16] Limited measurement of implementation fidelity [2,17]

**Table 2 children-09-01284-t002:** Health Equity Adapted Implementation Framework for Child Health Literacy Interventions.

Adapted CFIR Domains	Illustrative Questions
Characteristics of interventions	What evidence has informed the equitable implementation of child health literacy intervention to be tailored to local context and recipients?
Child–educator interaction	What perceptions and behaviours pertaining to disparities exist during child–educator interactions that may influence the equitable implementation of child health literacy interventions?
Culturally relevant factors (Recipients—children, parents, school staff)	What child and parent motivations, knowledge and beliefs may influence the receptivity and uptake of equitable implementation of child health literacy interventions? What school staff factors (e.g., knowledge, attitudes, beliefs and skillsets) may influence the equitable implementation of child health literacy interventions?
Inner context (local school organisation)	What local school organisational structures, networks and culture may influence the implementation of child health literacy interventions?
Outer context (education system)	What external education policies and incentives need consideration to support the equitable implementation of child health literacy interventions?
Societal context (economic, demographic, or geographical factors)	What costs or expenses may perpetuate health disparities in child health literacy? What built environment features within schools may influence the equitable implementation of child health literacy interventions? What policies or guidelines may promote or inhibit equitable implementation of child health literacy interventions?

## Data Availability

Not applicable.

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
