# Peer review of "A Health Equity Implementation Approach to Child Health Literacy Interventions"

_children, 2022, doi:10.3390/children9091284_

Round 1

Reviewer 1 Report

Generally, a well-written commentary on Child Health Literacy Interventions. However, even though the Manuscript is free from any glaring errors, it could benefit from English language editing.

Reviewer 2 Report

Abstract

L5. ‘Children’ isn’t a population. Do you mean “among populations of children”?

L6 The assertion “health literacy is a crucial driver to equitable health outcomes in children” is not clearly evidence-based and overstated.

L9 very confusing sentence that goes in several directions.

L11 If all populations were exactly the same we would expect same targets, processes and outcomes. They are not, so this sentence makes no sense.

L14 This final sentence introduces a “health equity implementation approach” but the reader can't be sure what this is yet. The sentence also implies health literacy causes inequalities. Very hard to follow.

 Body 

L22 Reference 1 and 2 provide no evidence to back up the strong assertion in the first line.

L27 This observation could make sense if it was framed around… because Children HL interventions are in their infancy, there is only a few disparate studies done and there are no clear directions.

L65 Seems that a major emphasis on implementation science is premature. Interventions and policies are frequently developed that don’t meet the needs of the target groups, and are not implementable. Using a strong implementation framework, even with a focus on inequity, when an intervention is already weak and easy to use by average or better resources children, schools or parents, will not only promulgate inequity, but cause it. Surely the emphasis should be on developing implementable interventions, which is done through the global movement of co-design with the end users, i.e., the children and teachers. Reference 7 Nash et al seem to be focused on this. Building equity into implementation is too late, it needs to be part of the initial design.

This paper has 3 sets of lists from reports or other papers (including a Table) with an inadequate explanations of them, inadequate definitions and explanations of what the issues and problems are, in what contexts, and to explain the root cause of the issue. 

Reviewer 3 Report

Your commentary “A Health Equity Implementation Approach to Child Health Literacy Interventions” addresses a very important, neglected topic that needs more attention in research, policy, and practice of health literacy. The “Health Equity Adapted Implementation Framework for Child Health Literacy Interventions” is a promising approach to bringing inequity reduction to health literacy promotion practice. THowever, the manuscript needs more clarity in some places so that the reader can better understand the rationale.

Some more comments:

The abstract should already present the proposed framework "Health Equity Adapted Implementation Framework for Child Health Literacy Interventions" with its features; prefer to shorten others if character count does not allow.

Provide some several references for the strong statement of line 52 "calls have emerged for a health equity implementation approach."

It remains unclear why promoting children's health literacy relates to implementation research in health care. In the preceding paragraphs, approaches were presented that do not relate to health care, but rather to schools and other educational settings. Please justify this in the paragraph line 53-71. In doing so, also check the associated heading, which refers to “Health Equity and Implementation” in general, but in the text the specification is made to "healthcare implementation".

It only becomes clear later how the content of lines 79-85 (the three types of frameworks) relates to your “Health Equity Adapted Implementation Framework for Child Health Literacy Interventions”. In this section, explain to the reader why you are presenting this in relation to the later part of the text (“Health Equity Adapted Implementation Framework for Child Health Literacy Interventions”). 

In addition to your general introduction to implementation frameworks, a more detailed explanation of the “Consolidated Framework for Implementation Research” is needed, as it is the focus of this paper. This includes: Why it is so widespread? Who is using it for what? Is it being used in many countries? Is it being used in health care? Is it already being used for interventions in education, especially schools?

If you would use statements instead of enumerations in the headings, it would be easier to follow the logical structure of the text. Try to apply this.

Line 19: Correct the headline from Manuscript into Introduction or similar. 

In line 33/34, one word is repeated. 

The statement in lines 36-38 refers to the World Health Organization, but the Reference cited is different. Please correct this. 

The enumeration lines should be removed from table 1. 

Throughout the table, lowercase/capitalization should be unified, as well as italics/non-italics in column 1 

I recommend to show the enumeration from line 40-52 with indents rather than with one or two line paragraphs. The same applies to the lines 76-86.

Reviewer 4 Report

Thank you very much for the opportunity to review this commentary entitled “A Health Equity Implementation Approach to Child Health Literacy Interventions.” Before it can be considered for publication in this journal, please address the following comments and suggestions:

General comments: 

This commentary lacks a global perspective. Readers will be interested in the disparities in child health literacy between high-income and low- and middle-income countries. It would help if the authors provided the current situation. Providing examples of successful and failed child health literacy interventions will also help to identify the research gaps and lessons learned from all these interventions. It is essential to focus first on these first two general comments before authors can transition to the health equity implementation approach.

Specific comments: 

L20-22: Provide specific examples of how health literacy is crucial to equitable outcomes in children. Please elaborate. 

L22-23: Could you provide specific examples of child health literacy interventions and explain their mechanism and how they improved healthy behaviors? 

L128-166: References are not professionally done and are inconsistent in using upper- and lower-case letters.

Round 2

Reviewer 2 Report

Thank you for the opportunity to undertake a second review. The authors seem assume a large knowledge base of the reader therefore many terms and concepts remain unclear and will be unclear to most readers.

I have provided advice in my first review and below and the Editor in Chief will need to consider if the article is acceptable quality for the readership.

L29 this repeats what is already stated in first paragraph. The second sentence adds very little also.

L44 I don’t see that it is appropriate to criticise Health Promoting Schools for being weak in health literacy – as that wasn’t their mandate. You cant just blame a program for doing something poorly when it wasn’t designed to do it. Surely you would want to include strengths in such a table? Surely you would want to showcase some children’s HL programs that worked well in particular contexts and achieved particular outcomes. Perhaps ‘areas for children health literacy development’ is a better frame. I don’t think that this table helps / adds anything given the list starting L64.

L52 This sentence doesn’t make sense. HL interventions are profoundly different because of different contexts and different purposes. It is poor practice to criticise a program for not being specific for criteria beyond the scope of the initial research. How can they have “implementation specificity” (a concept that doesn’t seem to make sense either, and is not defined/developed). This paper needs to frame the children health literacy field as being in early stages, has been undertaken in a limited range of settings, and has generated valuable impacts in some settings, but not others – as is the case in a developing field. Indeed, it could be enhanced if researchers applied implementation science after a strong program of work to get the projects/interventions right and evaluated and found to be worthy of implementation.

L78 Check this definition and provide a reference. The word ‘exists’ is a bit odd and implies equity be is throughout the setting (were socially just opportunities exist). There are many ways equity will not exist when there is opportunity and conditions to promote it.

L109 as will my initial review, I asked what was the value of the multiple lists of ideas presented in this short paper – without analysis or putting them in context. This list is presented in isolation, doesn’t seem to provide any insights, and isn’t referred to. I don’t think it adds.

L135 [apologies – I should have picked this up in first review] two references of practical examples of interventions seemed to be proposed. One is a protocol – this should be updated with the outcomes evaluation of the research cited. The other reference doesn’t have a proper reference and doesn’t seems to be a program published or with evidence. I suggest the cite a well formulated and evidence-based primary study, eg Nsangi et al Effects of the Informed Health Choices primary school intervention on the ability of children in Uganda to assess the reliability of claims about treatment effects: a cluster-randomised controlled trial. The Lancet 390(10092)pp374-388

L155 as stated in my first review and above, a program needs to be suitable to a particular context before implementation. Implementation Science is conditional on having a program worthy of implementation. This is my understanding based on 30 years experience, and I would expect my readers will understand when the see the term implementation. Even the application best practice equity-informed child-health literacy-informed implementation science to a program poorly designed for privileged kids will lead to inequality and implementation failure. 

Reviewer 3 Report

The Commentary has gained considerable clarity as a result of the revision. However, two points remain unclear:

Your answer “We thank the reviewer for this comment. Both authors acknowledge that children’s health can be promoted in multiple settings including: health care and schools and other educational settings. This Perspective is focused on schools and other educational settings as one of the settings and as an example where a health equity approach can be used- hence, the heading ‘Health Equity and Implementation ‘ is still appropriate  y comment “ We probably misunderstood each other. I also think that the education approach is the optimal one. But my point was that for your deduction in the commentary, you refer to findings/frameworks from the health care sector (“implementation research in health care”), although you are aiming at the education sector. I miss a sentence justifying this transfer. In doing so, also check the associated heading, which refers to “Health Equity and Implementation” in general, but in the text the specification is made to "healthcare implementation".

I hope I didn't miss it, but why didn't you address the following comment "If you used statements instead of bulleted lists in the headings, it would be easier to follow the logical structure of the text. Try to apply this." Please provide a follow up on this. 

Minor formal inaccuracies should still be corrected:

- Line 25, 27, 38 and 117 as well as in Table 1 the literature references should be written in one bracket

- Line 49: Capitalization of the first letter in "interventions"

- in Tab. 1: Missing space in the right column

- References: No. 4, 5, 6, 14. Correct author citations (compare to No. 7)

- References: Harmonize lower case/upper case letters of article titles and regard to a consistent upper case letters of the journal titles:

- References: Harmonize journal articles: either write out all or abbreviate all
